# Investigating Bias in Multilingual Language Models: Cross-Lingual Transfer of Debiasing Techniques

**Manon Reusens[1], Philipp Borchert[1,2], Margot Mieskes[3],**
**Jochen De Weerdt[1], Bart Baesens[1,4]**
[1]Research Centre for Information Systems Engineering (LIRIS), KU Leuven
[2]IESEG School of Management, 3 Rue de la Digue, 59000 Lille, France
[3]University of Applied Sciences Darmstadt
[4]Department of Decision Analytics and Risk, University of Southampton
{manon.reusens, philipp.borchert, jochen.deweerdt, bart.baesens}@kuleuven.be
margot.mieskes@h-da.de

DISCLAIMER: This paper contains explicit statements that are potentially offensive.

## Abstract

This paper investigates the transferability of debiasing techniques across different languages within multilingual models. We examine the applicability of these techniques in English, French, German, and Dutch. Using multilingual BERT (mBERT), we demonstrate that cross-lingual transfer of debiasing techniques is not only feasible but also yields promising results. Surprisingly, our findings reveal no performance disadvantages when applying these techniques to non-English languages. Using translations of the CrowS-Pairs dataset, our analysis identifies SentenceDebias as the best technique across different languages, reducing bias in mBERT by an average of 13%. We also find that debiasing techniques with additional pretraining exhibit enhanced cross-lingual effectiveness for the languages included in the analyses, particularly in lower-resource languages. These novel insights contribute to a deeper understanding of bias mitigation in multilingual language models and provide practical guidance for debiasing techniques in different language contexts.

## 1 Introduction

There has been a growing interest in addressing bias detection and mitigation in Natural Language Processing (NLP) due to their societal implications. Initially, research focused on debiasing word embeddings (Bolukbasi et al., 2016; Zhao et al., 2018b), but recent studies found that pretrained language models also capture social biases present in training data (Meade et al., 2022). Hence, attention has shifted towards debiasing techniques that target sentence representations. These techniques include additional pretraining steps (Zhao et al., 2019; Webster et al., 2020; Zmigrod et al., 2019)

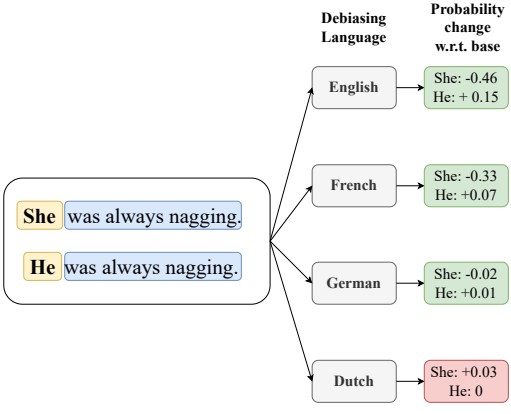

Figure 1: The example of the English CrowS-Pairs dataset illustrates sentence probabilities after debiasing mBERT with SentenceDebias in English, French, German, and Dutch.

and projection-based methods that assume a bias direction (Liang et al., 2020a; Ravfogel et al., 2020; Liang et al., 2020b).

While debiasing techniques have been developed and evaluated for monolingual, and mostly English models, the effectiveness and transferability of these techniques to diverse languages within multilingual models remain largely unexplored (Stanczak and Augenstein, 2021; Sun et al., 2019). Our research aims to bridge this gap by examining the potential of debiasing techniques applied to one language to effectively mitigate bias in other languages within multilingual large language models. We examine English (EN), French (FR), German (DE), and Dutch (NL). Figure 1 illustrates an example sentence pair included in the English CrowS-Pairs dataset [1], where the unmodified and modified parts are highlighted in blue and yellow respectively. It shows the predicted probabilities of

---

[1]This example assumes gender to be binary. We acknowledge that this fails to capture the full range of gender identities.

the modified part occurring given the unmodified part across different debiasing languages.

This study examines the cross-lingual transferability of debiasing techniques using mBERT. mBERT, trained on Wikipedia data from diverse languages, possesses the capability to process and generate text in various linguistic contexts. Despite balancing efforts, it still performs worse on low-resource languages (Wu and Dredze, 2020; Devlin, 2018). We investigate whether this performance disparity extends to gender, religious, and racial biases. Related studies demonstrate the effectiveness of cross-lingual debiasing for individual techniques and selected bias scopes (Liang et al., 2020b; Lauscher et al., 2021). We show how to reduce bias in mBERT across different languages by conducting a benchmark of state-of-the-art (SOTA) debiasing techniques and providing guidance on its implementation. To facilitate further research and reproducibility, we make the code and additional data available to the research community[2].

Our contributions can be summarized as follows: 1) We provide a benchmark of different SOTA debiasing techniques across multiple languages in a multilingual large language model. 2) We find that SentenceDebias is the most effective for cross-lingual debiasing, reducing the bias in mBERT by 13%. 3) We provide implementation guidelines for debiasing multilingual models and highlight the differences in the cross-lingual transferability of different debiasing techniques. We find that most projection-based techniques applied to one language yield similar predictions across evaluation languages. We also recommend performing the techniques with an additional pretraining step on the lowest resource language within the multilingual model for optimal results.

## 2 Methodology

This section introduces the data, debiasing techniques, and experimental setup respectively.

### 2.1 CrowS-Pairs

CrowS-Pairs is a benchmark dataset comprising 1508 examples that address stereotypes associated with historically disadvantaged groups in the US, encompassing various types of bias, such as age and religion (Nangia et al., 2020). Following Meade et al. (2022), where different debiasing tech-

---

[2] https://github.com/manon-reusens/multilingual_bias

niques were benchmarked and their effectiveness demonstrated on BERT for gender, race, and religion, we focus on these three types of bias. Névéol et al. (2022) translated the dataset in French. To the best of our knowledge, there are currently no peer-reviewed variants of CrowS-Pairs available in other languages. Therefore, we used three samples of the full dataset and translated them into the respective language to evaluate our experiments.

To create an evaluation set for our experiments, we started from the English CrowS-Pairs dataset (Nangia et al., 2020). We randomly sampled $N$ instances, where $N \in \{20, 30, 40, 50\}$, and measured the performance differences on mBERT and BERT. Through three random seeds, we found that a sample size of 40 resulted in an average performance correlation of more than 75% with the full dataset for both models. Thus, we conclude that using 40 instances with three random seeds provides a representative dataset for our evaluation. Further details are shown in Appendix A. Subsequently, we included the translated samples from each language into our dataset, either the corresponding sentences from the French CrowS-Pairs or a translation.

### 2.2 Debiasing techniques

Next, the different debiasing techniques are explained. For more information on the attribute lists used, we refer to Appendix B.

**Counterfactual Data Augmentation (CDA)** is a debiasing technique that trains the model on an augmented training set (Zhao et al., 2019; Webster et al., 2020; Zmigrod et al., 2019). First, the corpus is augmented by duplicating sentences that include words from a predefined attribute list. Next, counterfactual sentences are generated by swapping these attribute words with other variants in the list, for example, swapping *he* by *she*. We augment 10% of the Wikipedia corpus of the respective language and use an additional pretraining step to debias the model for three random seeds and average the results.

**Dropout Regularization (DO)** is introduced by Webster et al. (2020) as a debiasing technique by implementing an additional pretraining step. We execute this pretraining step while training on 10% of the Wikipedia corpus of the respective language using three random seeds and averaging the results.

**SentenceDebias (SenDeb)** introduced by Liang et al. (2020a) is a projection-based debiasing technique extending debiasing word embeddings

(Bolukbasi et al., 2016) to sentence representations. Attribute words from a predefined list are contextualized by retrieving their occurrences from a corpus and augmented with CDA. Next, the bias subspace is computed using the representations of these sentences through principal component analysis (PCA). The first $K$ dimensions of PCA are assumed to define the bias subspace as they capture the principle directions of variation of the representations. We debias the last hidden state of the mBERT model and implement SenDeb using 2.5% of the Wikipedia text in the respective language.

**Iterative Nullspace Projection (INLP)** is a projection-based debiasing technique in which multiple linear classifiers are trained to predict biases, such as gender, that are to be removed from the sentence representations (Ravfogel et al., 2020). After training a single classifier, the representations are debiased by projecting them onto the learned linear classifier's weight matrix to gather the rowspace projection. We implement this technique using the 2.5% of the Wikipedia text in each language.

**DensRay (DR)** is a projection-based debiasing technique first implemented by (Dufter and Schütze, 2019) and adapted for contextualized word embeddings by (Liang et al., 2020b). This technique is similar to SenDeb, but the bias direction is calculated differently. This method aims to find an optimal orthogonal matrix so that the first $K$ dimensions correlate well with the linguistic features in the rotated space. The second dimension is assumed to be orthogonal to the first one. The bias direction is considered to correspond to the eigenvector of the highest eigenvalue of the matrix. DR is only implemented for a binary bias type and using it for multiclass bias types requires modifying the technique. Therefore, we only apply it to the gender bias type. We implement DR debiasing the last hidden state of mBERT and using 2.5% of the Wikipedia text in the respective language.

## 2.3 Experimental setup

We debias mBERT using language $X$ and evaluating it on language $Y$ with $X, Y \in \{EN, FR, DE, NL\}$. In essence, we debiased the model using one language and evaluated it on another, covering all language combinations in our experiments. We implement mBERT in its base configuration (uncased, 12 layers, 768 hidden size) and utilize the bias score as implemented in Meade et al. (2022). This metric evaluates the percentage of sentences where the model prefers the more biased sentence over the less biased sentence, with an optimal performance of 50%. All experiments are performed on P100 GPUs.

## 3 Results

Table 1 shows the performance of the different debiasing techniques when debiasing in English in terms of the absolute deviation of the ideal unbiased model. This is an average score for all bias types and models trained for the respective evaluation language. Base represents the score that is achieved by mBERT on the respective evaluation language dataset before debiasing. More results are shown in Appendices C and D.

As shown in Table 1, English is relatively unbiased compared to the other languages and shows a small bias increase after debiasing. This observation aligns with the findings of Ahn and Oh (2021), who propose mBERT as a debiasing technique. In cases where the initial bias score is already close to the optimal level, further debiasing can lead to *overcompensation*, consequently amplifying the total bias. We assume that an unbiased model should equally prioritize both biased and unbiased sentences. However, when debiasing techniques tend to overcorrect, they skew the balance towards favoring the prediction of unbiased sentences over biased ones. Addressing this challenge necessitates the adoption of specialized techniques to effectively mitigate any residual bias.

This phenomenon of overcompensation occurs in several underperforming techniques, as illustrated in Table 1. Notably, we find instances of overcompensation for gender when debiasing using INLP for French and using CDA for German, as well as for race when debiasing using DO for German. Another contributing factor to the poor performance of certain techniques within specific debiasing and evaluation language combinations lies in the inherent ineffectiveness of the debiasing method itself, exemplified by the cases of gender debiasing using CDA for French and religion debiasing using CDA for German. In Tables 5, 6, and 7, we find overcompensation for gender when debiasing with INLP in German and French, evaluating in German, debiasing with Sendeb and DR in French, and evaluating in French, as well as when debiasing in Dutch with INLP and evaluating in French. Moreover, overcompensation for race is also observed when debiasing with CDA in French

| | Base | INLP | Sendeb | DR | CDA | DO |
|---|---|---|---|---|---|---|
| EN | 6.11 | 8.70↑ | 7.78↑ | 6.94↑ | 13.43↑ | 8.70↑ |
| FR | 11.11 | 11.20↑ | 10↓ | 10.28↓ | 12.6↑ | 9.44↓ |
| DE | 9.33 | 7.52↓ | 6.57↓ | 6.84↓ | 10.75↑ | 9.75↑ |
| NL | 17.66 | 13.96↓ | 15.14↓ | 16.54↓ | 16.84↓ | 17.40↓ |

Table 1: Overall performance score per evaluation language and debiasing technique averaged over the three random seeds after debiasing in English.

| Evaluation language | Best debiasing language | worst debiasing language |
|---|---|---|
| English | Dutch | French |
| French | German | French |
| German | German | French |
| Dutch | French | English |

Table 2: Overview best- and worst-performing debiasing languages per evaluation language.

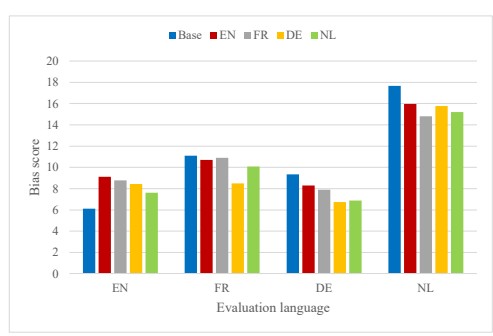

Figure 2: Average bias scores per evaluation and debiasing language.

and evaluating in German.

**Is cross-lingual transferability of debiasing techniques possible?** Table 1 shows that cross-lingual transfer is possible using English as debiasing language. Figure 2 confirms this, depicting the bias scores averaged over all debiasing techniques. As discussed, for English, these techniques increase the bias contained in the model due to its already close to optimal performance. For the other evaluation languages, we find better performance after debiasing. Therefore, we conclude that for these four languages, it is possible to debias the mBERT model to some extent using a different debiasing language, except when the bias contained in the model is already relatively low.

To shed some light on the insights that can be gathered from Figure 2, Table 2 offers an overview of the best- and worst-performing techniques per evaluation language. As shown, Dutch is found to be the best debiasing language for English. This is because this debiasing language has shown to overcompensate the gender bias category the least, therefore, resulting in the best performance. In general, we find that using the same debiasing language as evaluation language often results in an overcompensation of the bias, therefore turning around the bias direction. This means that the best-performing debiasing language is often not the same as the evaluation language. However, German is the exception. As this language already has the highest bias score for gender before debiasing, strong debiasing is beneficial and therefore does not result in overcompensation. Besides German being the best-performing debiasing language for German, it also shows the best performance for French because it achieves the best performance on all different evaluation sets. Moreover, it also shows less overcompensation for the gender bias present in the model than other languages such as Dutch.

French is the worst-performing debiasing language for all evaluation languages except for Dutch, where it is the best-performing one. We find that when evaluating in French, the gender bias is over-compensated. For English, both racial and gender bias are overcompensated. The German evaluation shows lower overall performance due to already two ineffective methods (INLP and CDA), which were also due to overcompensating racial bias. Finally, for Dutch, we find that debiasing with French overcompensates gender bias less than Dutch and, therefore, is the best-performing method. As Dutch has the second highest gender bias score before debiasing, it also benefits from strong debiasing and therefore both French and Dutch perform well.

We believe that these results are influenced by the fact that both German and French have a grammatical gender distinction, which may impact debiasing gender to a greater extent. This grammatical gender distinction is not embedded in English and Dutch. Moreover, as the religion category regularly shows limited bias decrease, we find that the performance in the gender and race category often determines whether a technique works well or not.

**How are the different techniques affected by cross-lingual debiasing?** Table 3 shows the overall percentage increase of the bias score per technique. From this, we conclude that SenDeb is the best-performing technique and reduces bias in mBERT on average by 13%. DO is the second best-performing method reducing bias on average by 10%. However, Figure 3 shows that DO performs well for all debiasing languages except English, while SenDeb performs consistently well for all languages. The other techniques perform worse overall. Hence, we suggest using SenDeb as cross-lingual debiasing technique for these languages.

| INLP | Sendeb | DR | CDA | Dropout |
|------|--------|-----|-------|---------|
| 1.11 | 13.21 | 7.1 | -0.16 | 10.21 |

Table 3: Average percentage difference in bias scores of each technique compared to the base model.

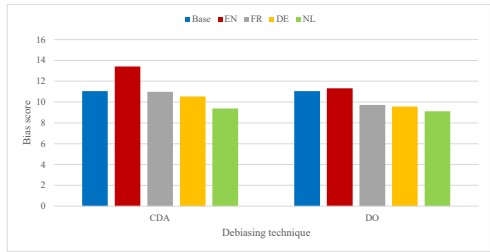

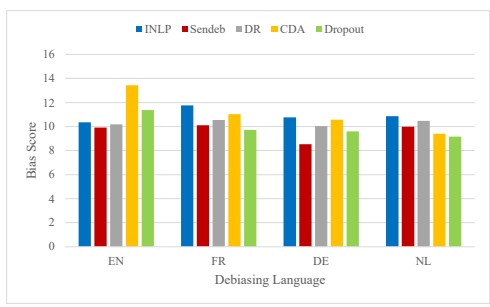

Figure 3: Average bias score per technique per debiasing language.

When zooming in on the **projection-based techniques**, i.e. INLP, SenDeb, and DR, a high performance variation is shown in Table 3 and Figure 3. While SenDeb offers consistent performance for all different debiasing languages, we see more variation and a lower bias decrease for INLP. This is due to the high variation in performance, resulting in a higher overall average. As INLP uses multiple linear classifiers to define the projection matrix, high variability is introduced. Since DR was only implemented for gender, no performance gains can be obtained from the other bias types, therefore resulting in a lower overall performance increase.

Techniques using an **additional pretraining step** obtain the best results when debiasing in Dutch, as illustrated in Figure 4. Notably, Dutch is the lowest resource language out of these four languages during pretraining (Wu and Dredze, 2020). This additional pretraining step lets the model learn unbiased associations between words while becoming familiar with the lower-resource language resulting in lower overall bias. Therefore, we conclude that, for our set of languages, these techniques are most effective when applied to low-resource languages.

## 4   Related work

Significant research focuses on the cross-lingual performance of mBERT (Wu and Dredze, 2020; Pires et al., 2019; Libovický et al., 2019). Limited research focuses on the cross-lingual transferability of debiasing techniques in mBERT (Stanczak and Augenstein, 2021; Sun et al., 2019). Liang et al. (2020b) use DensRay in English to debias Chinese

Figure 4: Average bias score per debiasing languages for both CDA and DO.

in mBERT for gender. Similarly, Zhao et al. (2020) analyze the cross-lingual transfer of gender bias mitigation using one method. Lauscher et al. (2021) also find that their proposed technique, ADELE, can transfer debiasing across six languages. Other studies analyze biases contained in multilingual language models. Kaneko et al. (2022) evaluate bias across multiple languages in masked language models using a new metric. Ahn and Oh (2021) study ethnic bias and its variability over languages proposing mBERT as debiasing technique. Finally, some studies also explore the cross-lingual transferability of downstream tasks (Levy et al., 2023).

## 5   Conclusion

Most studies focus on debiasing techniques for large language models, but rarely explore their cross-lingual transferability. Therefore, we offer a benchmark for SOTA debiasing techniques on mBERT across multiple languages (EN, FR, DE, NL) and show that debiasing is transferable across languages, yielding promising results. We provide guidance for cross-lingual debiasing, highlighting SenDeb as the best-performing method, reducing bias in mBERT by 13%. Additionally, we find that, for the studied languages, debiasing with the lowest resource language is effective for techniques involving an additional training step (CDA and DO). This research is a first step into the cross-lingual transferability of debiasing techniques. Further studies should include languages from different cultures and other multilingual large language models to assess generalizability.

## Limitations

A first limitation concerns the analysis focused on four closely related languages from a similar culture. A broader range of languages should be explored to ensure the generalizability of the findings. Our research was conducted employing a

single multilingual model, mBERT. Extending this to other multilingual language models would provide valuable insights into the wider applicability of the results. Moreover, the evaluation of outcomes relied primarily on the CrowS-Pairs metric, although efforts were made to enhance the understanding by examining the absolute difference compared to the optimal model. Next, the consideration of gender was limited to binary classification, overlooking non-binary gender identities. This should be further addressed in future research. Furthermore, a comprehensive multilingual dataset to assess stereotypes across different languages is not available, and thus, the English CrowS-Pairs dataset was translated and corresponding sentences of the French dataset were used. Nevertheless, typical stereotypes for other languages were not adequately represented. Furthermore, the dataset used in the study exhibited certain flaws highlighted by Blodgett et al. (2021), such as the influence of selected names on predictions, which was observed to have a significant impact. This needs to be investigated further. Additionally, attribute lists for languages other than English were not available to the same extent. We tried to compile lists for French, Dutch, and German, excluding words with multiple meanings to minimize noise. However, our lists were not exhaustive, and therefore the omission of relevant attributes is possible. It is also worth noting that, in certain cases, the generic masculine form was considered the preferred answer, despite it being included in the attribute lists. Finally, the applicability of downstream tasks should be investigated (e.g. (Levy et al., 2023)). Hence, future research should encompass a wider language scope, include multiple models, address existing dataset flaws, and develop more comprehensive attribute lists for various languages.

## Ethics Statement

We would like to address three key ethical considerations in this study that highlight ongoing challenges and complexities associated with mitigating bias in large language models. First, it is important to acknowledge that the gender bias examined in this paper is approached from a binary perspective. However, this does not capture the full range of gender identities present in reality. While we recognize this limitation, it was necessary to simplify the analysis for experimental purposes. In future research, we hope to address this limitation. Second, despite

efforts to debias the multilingual large language model, it is important to note that not all forms of bias are completely mitigated. The applied debiasing techniques do lower the bias present in the model, however, there is still bias present in the model both within and outside the targeted bias types. Finally, we recognize that our evaluation datasets do not encompass all the different biases that might be present in the model. Therefore, even if a model would obtain a perfect score, it is still possible that other forms of bias persist.

## Acknowledgements

This research was funded by the Statistics Flanders research cooperation agreement on Data Science for Official Statistics. The resources and services used in this work were provided by the VSC (Flemish Supercomputer Center).

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

## A    Correlation samples and full dataset

Table 4 shows the overall correlation between the performance of the respective model on the full dataset and on the sample over three random seeds. Full represents that we looked at all different bias types in the CrowS-Pairs dataset and (3) refers to the three bias types used in this study (gender, religion, and race). We used a threshold of 75% to decide on the sample size.

|         | mBERT full | mBERT (3) | BERT full | BERT (3) |
|---------|------------|-----------|-----------|----------|
| size 20 | 85.71      | 66.58     | 74.40     | 70.58    |
| size 30 | 88.54      | 66.80     | 80.14     | 70.66    |
| size 40 | 93.11      | 86.02     | 83.78     | 75.88    |
| size 50 | 94.21      | 86.82     | 90.52     | 87.32    |

Table 4: Correlation between the sample and full respective model's performance over three random seeds before debiasing.

## B    Attribute Lists

The different lists of attribute words per language used for CDA, SenDeb, INLP, and DR are listed below. We started from the lists in  (Meade et al., 2022; Zhao et al., 2018a; Liang et al., 2020a) and extended these to other languages. For the non-English lists, we first translated the words in the English attribute list if a similar relation is contained in the translated words. Subsequently, we asked a native speaker of the respective language whether there are extra words that contain similar relations which do not exist in English. We added these to the list. However, note that these lists are not exhaustive, as many other words still exist that express the same relationship. Nevertheless, this list already provides a more extensive attribute list than by only translating the English attribute list.

### English

**Gender**:(Zhao et al., 2018a) *actor, actress, actors, actresses, airman, airwoman, airmen, airwomen, uncle, aunt, uncles, aunts, boy, girl, boys, girls, groom, bride, grooms, brides, brother, sister, brothers, sisters, businessman, businesswoman, businessmen, businesswomen, chairman, chairwoman, chairmen, chairwomen, dude, chick, dudes, chicks, dad, mom, dads, moms, daddy, mommy, daddies, mommies, son, daughter, sons, daughters, father, mother, fathers, mothers, male, female, males, females, guy, gal, guys, gals, gentleman, lady, gentlemen, ladies, grandson, granddaughter, grandsons, granddaughters, guy, girl, guys, girls, he, she, himself, herself, him, her, his, her, husband, wife, husbands, wives, king, queen, kings, queens, lord, lady, lords, ladies, sir, ma'am, man, woman, men, women, sir, miss, mr., mrs., mr., ms., policeman, policewoman, prince, princess, princes, princesses, spokesman, spokeswoman, spokesmen, spokeswomen.*
**Race**: (Meade et al., 2022) *black, caucasian, asian, african, caucasian, asian, black, white, asian, africa, america, asia, africa, america, china, africa, europe, asia.*
**Religion**: (Liang et al., 2020a) *jewish, christian, muslim, jews, christians, muslims, torah, bible, quran, synagogue, church, mosque, rabbi, priest, imam, judaism, christianity, islam.*

### French

**Gender**: *acteur, actrice, acteurs, actrices, aviateur, aviatrice, aviateurs, aviatrices, oncle, tante, oncle, tantine, tonton, tata, tontons, tatas, oncles, tantes, oncles, tantines, garçon, fille, garçons, filles, marié, mariée, époux, épouse, mariés, mariées, époux, épouses, brother, sister, frère, soeur, frères, soeurs, entrepreneur, entrepreneuse, entrepreneur, entrepreneure, entrepreneurs, entrepreneures, entrepreneurs, entrepreneuses, président, présidente, présidents, présidentes, mec, meuf, gamin, gamine, mecs, meufs, père, mère, pères, mères, papa, maman, papas, mamans, fils, fille, fils, filles, abbé, abbesse, abbés, abbesses, masculin, féminin, mâle, femelle, mâles, femelles, gars, fille, gars, filles, monsieur, dame, messieurs, dames, petit-fils, petite-fille, petit-fils, petites-filles, il, elle, lui-même, elle-même, lui, elle, mari, femme, maris, femmes, roi, reine, rois, reines, seigneur, seigneuresse, seigneurs, seigneuresses, monsieur, m'dame, monsieur, madame, homme, femme, hommes, femmes, monsieur, mademoiselle, mr, mme, mr, mlle, policier, policière, prince,*

*princesse, princes, princesses, copain, copine, co-pains, copines, ami, amie, amis, amies, voisin, voisine, docteur, doctoresse, docteur, docteure, boulanger, boulangère, héros, héroïne, employé, employée, employés, employées, chef, cheffe, chefs, cheffes, cousin, cousine, grand-père, grand-mère, expert, experte, pompier, pompière, pompiers, pompières, agriculteur, agricultrice, agriculteurs, agricultrices, travailleur, travailleuse, infirmier, infirmière, infirmiers, infirmières, patron, patronne, patrons, patronnes.*

**Race**: *noir, blanc, asiatique, black, blanc, asiatique, noir, caucasien, asiatique, africain, européen, asienne, africain, américain, asiatique, afrique, amérique, asie, afrique, amérique, chine, afrique, europe, asie.*

**Religion**: *juif, chrétien, musulman, juifs, chrétiens, musulmans, torah, bible, coran, synagogue, église, mosquée, rabbin, prêtre, imam, judaïsme, christianisme, islam.*

### German

**Gender:** *schauspieler, schauspielerin, koch, köchin, lehrer, lehrerin, schüler, schülerin, student, studentin, pilot, pilotin, onkel, tante, junge, mädchen, bräutigam, braut, bruder, schwester, geschäftsmann, geschäftsfrau, vorsitzender, vorsitzende, vater, mutter, papa, mama, sohn, tochter, mann, frau, kerl, mädel, herr, dame, enkel, enkelin, großvater, großmutter, cousin, cousine, er, sie, ihm, ihr, sein, ihr, seine, ihre, ehemann, ehefrau, feuerwehrmann, feuerwehrfrau, könig, königin, fürst, fürstin, herzog, herzogin, mann, frau, männer, frauen, hr., fr., polizist, polizistin, prinz, prinzessin, sprecher, sprecherin, kollege, kollegin, mitarbeiter, mitarbeiterin, helfer, helferin, anwalt, anwältin, bauarbeiter, bauarbeiterin, krankenpfleger, krankenpflegerin, chef, chefin, vorgesetzter, vorgesetzte, sänger, sängerin, kunde, kundin, besucher, besucherin, freund, freundin, arzt, ärztin, verkäufer, verkäuferin, kanzler, kanzlerin, geschäftsleiter, geschäftsleiterin, pfleger, pflegerin, kellner, kellnerin.*

**Race:** *dunkelhäutig, hellhäutig, asiatisch, afrikaner, europäer, asiate, amerikaner, afrika, amerika, asien, china.*

**Religion:** *jüdisch, christlich, muslimisch, jude, christ, muslim, torah, bibel, koran, synagoge, kirche, moschee, rabbiner, pfarrer, imam, judentum, christentum, islam.*

### Dutch

**Gender:** *acteur, actrice, acteurs, actrices,*

*oom, tante, ooms, tantes, nonkel, tante, nonkels, tantes, jongen, meisje, jongens, meisjes, bruidegom, bruid, bruidegommen, bruiden, broer, zus, broers, zussen, zakenman, zakenvrouw, zakenmannen, zakenvrouwen, kerel, griet, kerels, grieten, vader, moeder, vaders, moeders, papa, mama, papa's, mama's, zoon, dochter, zonen, dochters, man, vrouw, mannen, vrouwen, gast, wijf, gasten, wijven, heer, dame, heren, dames, kleinzoon, kleindochter, kleinzonen, kleindochters, vent, vrouw, venten, vrouwen, hij, zij, hemzelf, haarzelf, hem, haar, zijn, haar, mannelijk, vrouwelijk, vriend, vriendin, vrienden, vriendinnen, koning, koningin, koningen, koninginnen, heer, dame, heren, dames, meneer, mevrouw, jongeheer, jongedame, jongeheren, jongedames, jongeheer, juffrouw, jongeheren, juffrouwen, politieagent, politieagente, prins, prinses, prinsen, prinsessen, woordvoerder, woordvoerster, woordvoerders, woordvoersters, brandweerman, brandweervrouw, brandweermannen, brandweervrouwen, timmerman, timmervrouw, timmermannen, timmervrouwen, meester, juf, meesters, juffen, verpleger, verpleegster, verplegers, verpleegsters, bestuurder, bestuurster, bestuurders, bestuursters, kuisman, kuisvrouw, kuismannen, kuisvrouwen, kok, kokkin, kokken, kokkinnen, leraar, lerares, directeur, directrice, directeurs, directrices, secretaris, secretaresse, secretarissen, secretaressen, boer, boerin, boeren, boerinnen, held, heldin, gastheer, gastvrouw, gastheren, gastvrouwen, opa, oma, opa's, oma's, grootvader, grootmoeder, grootvaders, grootmoeders.*

**Race:** *afrikaans, amerikaans, aziatisch, afrikaans, europees, aziatisch, zwart, blank, aziatisch, afrika, amerika, azië, afrika, amerika, china, afrika, europa, azië.*

**Religion:** *joods, christen, moslim, joden, christenen, moslims, thora, bijbel, koran, synagoge, kerk, moskee, rabbijn, priester, imam, jodendom, christendom, islam.*

## C   Averaged results

Tables 5, 6, and 7 show the different averaged results are the debiasing languages other than English, namely French, German, and Dutch.

## D   Breakdown results

In this section, a breakdown of the different scores per category is shown in terms of the bias metric established in (Nangia et al., 2020) in Tables 8, 9, 10, 11. For brevity, we employ the following

|  | mBERT | INLP | SenDeb | DR* | CDA | DO |
|---|---|---|---|---|---|---|
| EN | 6.11 | 10.93 ↑ | 6.94 ↑ | 7.50 ↑ | 9.44 ↑ | 9.07 ↑ |
| FR | 11.11 | 9.91 ↓ | 12.22 ↑ | 11.67 ↓ | 10.00 ↓ | 10.74 ↓ |
| DE | 9.33 | 11.11 ↑ | 6.29 ↓ | 6.55 ↓ | 9.45 ↑ | 6.09 ↓ |
| NL | 17.66 | 14.96 ↓ | 14.86 ↓ | 16.26 ↓ | 15.05 ↓ | 12.94 ↓ |

Table 5: Overall performance score per evaluation language and debiasing technique averaged over the three random seeds after debiasing in French.

|  | mBERT | INLP | SenDeb | DR* | CDA | DO |
|---|---|---|---|---|---|---|
| EN | 6.11 | 9.44 ↑ | 6.94 ↑ | 7.22 ↑ | 10.19 ↑ | 8.43 ↑ |
| FR | 11.11 | 9.17 ↓ | 7.5 ↓ | 10.28 ↓ | 8.43 ↓ | 7.13 ↓ |
| DE | 9.33 | 10.20 ↑ | 4.89 ↓ | 6.27 ↓ | 6.38 ↓ | 6.01 ↓ |
| NL | 17.66 | 14.31 ↓ | 14.59 ↓ | 16.25 ↓ | 17.10 ↓ | 16.65 ↓ |

Table 6: Overall performance score per evaluation language and debiasing technique averaged over the three random seeds after debiasing in German.

|  | mBERT | INLP | SenDeb | DR* | CDA | DO |
|---|---|---|---|---|---|---|
| EN | 6.11 | 8.80 ↑ | 6.94 ↑ | 7.22 ↑ | 7.69 ↑ | 7.50 ↑ |
| FR | 11.11 | 11.30 ↑ | 10.28 ↓ | 10.56 ↓ | 9.07 ↓ | 9.17 ↓ |
| DE | 9.33 | 8.71 ↓ | 6.83 ↓ | 7.39 ↓ | 6.04 ↓ | 5.37 ↓ |
| NL | 17.66 | 14.68 ↓ | 15.71 ↓ | 16.54 ↓ | 14.69 ↓ | 14.41 ↓ |

Table 7: Overall performance score per evaluation language and debiasing technique averaged over the three random seeds after debiasing in Dutch.

abbreviations: Gender (G), Race (Ra), and Religion (Re).

|  | mBERT | INLP | SenDeb | DR* | CDA | DO |
|---|---|---|---|---|---|---|
| EN | 51.11 | 51.30 | 51.11 | 50.28* | 55.28 | 50.37 |
| G | 49.17 | 49.17 | 49.17 | 46.67 | 50.83 | 51.11 |
| Ra | 44.17 | 41.94 | 45.00 | - | 39.72 | 39.17 |
| Re | 60.00 | 62.78 | 59.17 | - | 75.28 | 60.83 |
| FR | 60.56 | 57.50 | 60.00 | 60.28* | 62.59 | 58.52 |
| G | 50.83 | 45.56 | 50.00 | 50.00 | 59.72 | 60.83 |
| Ra | 59.17 | 56.67 | 57.50 | - | 56.39 | 50.83 |
| Re | 71.67 | 70.28 | 72.50 | - | 71.67 | 63.89 |
| DE | 59.05 | 55.93 | 54.86 | 56.55* | 57.06 | 55.50 |
| G | 60.90 | 54.50 | 51.71 | 53.42 | 53.29 | 56.70 |
| Ra | 57.07 | 61.35 | 57.05 | - | 51.51 | 48.42 |
| Re | 59.17 | 51.94 | 55.83 | - | 66.39 | 61.39 |
| NL | 67.66 | 63.96 | 64.59 | 65.98* | 66.65 | 67.22 |
| G | 56.32 | 54.10 | 49.59 | 51.28 | 56.62 | 57.48 |
| Ra | 65.83 | 59.17 | 65.00 | - | 66.39 | 65.56 |
| Re | 80.83 | 78.61 | 79.17 | - | 76.94 | 78.61 |

Table 8: Overall performance score per evaluation language, debiasing technique, and category averaged over the three random seeds after debiasing in English.

|  | mBERT | INLP | SenDeb | DR* | CDA | DO |
|---|---|---|---|---|---|---|
| EN | 51.11 | 55.74 | 50.28 | 50.28* | 54.26 | 51.67 |
| G | 49.17 | 52.78 | 47.50 | 46.67 | 49.72 | 53.06 |
| Ra | 44.17 | 45.83 | 43.33 | - | 46.11 | 38.89 |
| Re | 60.00 | 68.61 | 60.00 | - | 66.94 | 63.06 |
| FR | 60.56 | 54.54 | 60.00 | 58.89* | 58.15 | 57.59 |
| G | 50.83 | 47.22 | 46.67 | 45.83 | 52.22 | 55.28 |
| Ra | 59.17 | 47.22 | 60.83 | - | 57.50 | 51.67 |
| Re | 71.67 | 69.17 | 72.50 | - | 64.72 | 65.83 |
| DE | 59.05 | 59.68 | 55.43 | 55.98* | 55.95 | 53.09 |
| G | 60.90 | 52.47 | 50.85 | 51.71 | 55.24 | 55.83 |
| Ra | 57.07 | 59.91 | 58.78 | - | 52.05 | 47.04 |
| Re | 59.17 | 66.67 | 56.67 | - | 60.56 | 56.39 |
| NL | 67.66 | 64.21 | 64.31 | 65.71* | 65.05 | 62.94 |
| G | 56.32 | 49.59 | 49.59 | 50.45 | 54.32 | 55.76 |
| Ra | 65.83 | 62.22 | 65.00 | - | 62.50 | 58.33 |
| Re | 80.83 | 80.83 | 78.33 | - | 78.33 | 74.72 |

Table 9: Overall performance score per evaluation language and debiasing technique averaged over the three random seeds after debiasing in French.

|  | mBERT | INLP | SenDeb | DR* | CDA | DO |
|---|---|---|---|---|---|---|
| EN | 51.11 | 54.44 | 50.28 | 50.00* | 52.59 | 52.13 |
| G | 49.17 | 46.67 | 45.83 | 45.83 | 45.28 | 49.17 |
| Ra | 44.17 | 53.06 | 45.00 | - | 44.17 | 43.06 |
| Re | 60.00 | 63.61 | 60.00 | - | 68.33 | 64.17 |
| FR | 60.56 | 56.57 | 57.5 | 60.28* | 53.80 | 53.61 |
| G | 50.83 | 47.48 | 50.00 | 50.00 | 53.89 | 55.83 |
| Ra | 59.17 | 60.56 | 58.33 | - | 44.44 | 45.83 |
| Re | 71.67 | 61.39 | 64.17 | - | 63.06 | 59.17 |
| DE | 59.05 | 57.64 | 53.75 | 55.70* | 55.72 | 55.35 |
| G | 60.90 | 47.69 | 50.00 | 50.86 | 54.46 | 58.41 |
| Ra | 57.07 | 64.38 | 57.07 | - | 57.69 | 53.47 |
| Re | 59.17 | 60.83 | 54.17 | - | 55.00 | 54.17 |
| NL | 67.66 | 61.16 | 63.48 | 65.14* | 67.10 | 66.65 |
| G | 56.32 | 49.88 | 48.76 | 48.76 | 56.30 | 56.60 |
| Ra | 65.83 | 53.89 | 64.17 | - | 69.44 | 67.78 |
| Re | 80.83 | 79.72 | 77.50 | - | 75.56 | 75.56 |

Table 10: Overall performance score per evaluation language and debiasing technique averaged over the three random seeds after debiasing in German.

|  | mBERT | INLP | SenDeb | DR* | CDA | DO |
|---|---|---|---|---|---|---|
| EN | 51.11 | 54.91 | 50.83 | 50.56* | 47.69 | 51.20 |
| G | 49.17 | 51.94 | 48.33 | 47.50 | 45.56 | 52.22 |
| Ra | 44.17 | 50.00 | 44.17 | - | 40 | 40.83 |
| Re | 60.00 | 62.78 | 60.00 | - | 57.5 | 60.56 |
| FR | 60.56 | 60.37 | 60.28 | 60.56* | 55.37 | 52.50 |
| G | 50.83 | 50.28 | 50.83 | 50.83 | 55.83 | 53.89 |
| Ra | 59.17 | 61.94 | 61.67 | - | 44.44 | 41.11 |
| Re | 71.67 | 68.89 | 68.33 | - | 65.83 | 62.50 |
| DE | 59.05 | 55.67 | 56.55 | 56.25* | 48.53 | 50.86 |
| G | 60.90 | 47.71 | 53.38 | 52.52 | 53.04 | 54.73 |
| Ra | 57.07 | 58.20 | 59.59 | - | 45.89 | 46.47 |
| Re | 59.17 | 61.11 | 56.67 | - | 46.67 | 51.39 |
| NL | 67.66 | 64.21 | 63.74 | 64.57* | 64.51 | 63.67 |
| G | 56.32 | 52.92 | 47.05 | 47.05 | 52.97 | 53.22 |
| Ra | 65.83 | 59.72 | 67.50 | - | 61.67 | 57.78 |
| Re | 80.83 | 80.00 | 76.67 | - | 78.89 | 80.00 |

Table 11: Overall performance score per evaluation language and debiasing technique averaged over the three random seeds after debiasing in Dutch.