# OpenReview forum: "Investigating Bias in Multilingual Language Models: Cross-Lingual Transfer of Debiasing Techniques"
_EMNLP/2023/Conference — EMNLP 2023 Main_

### Official Review · Reviewer_RwvA · 2023-07-30

**Typos Grammar Style And Presentation Improvements:** N/A
**Soundness:** 3

**Excitement:**

4: Strong: This paper deepens the understanding of some phenomenon or lowers the barriers to an existing research direction.

**Missing References:**

N/A

**Paper Topic And Main Contributions:**

This paper examines the effect of cross-lingual transfer on various debiasing techniques and finds, it is possible to perform debiasing while following the route of cross-lingual transfer. Moreover, the research identifies SentenceDebias as the best-performing debiasing technique in this specific setting. Additional findings include obtaining better results for lower-resourced languages.

**Questions For The Authors:**

Can you include a more diverse set of evaluation to prove the points the study makes or presents any argument that supports the reason of not including additional languages? Is these four languages enough to reach a conclusion about the general nature of cross-lingual transfer and debiasing, if yes, then why?

**Reasons To Accept:**

1. Covering a diverse list of debiasing techniques.
2. Straightforward and substantial result reporting based on the experiments the study covers.

**Reasons To Reject:**

The main problem, I feel like not having enough languages for evaluation to reach toward a conclusion. All 4 languages in this study are in latin script. What happens when we change this route and evaluate on a non-Latin script language? Also, to reach a conclusion about low-resource language, I feel more experiments need to be done on other true low-resource languages such as not seen by mbert languages.

**Reproducibility:**

5: Could easily reproduce the results.

**Reviewer Confidence:**

4: Quite sure. I tried to check the important points carefully. It's unlikely, though conceivable, that I missed something that should affect my ratings.

---

> ### Author Rebuttal · Authors · 2023-08-29
>
> We thank you for your time and valuable insights that contributed to the paper’s quality. We agree that this paper studies a diverse list of debiasing techniques and reports results regarding the numerous experiments we carried out in a straightforward manner. In the following paragraphs we elaborate on your feedback and remarks:
> >R3A: What happens when we change this route and evaluate on a non-Latin script language? R3C: Can you include a more diverse set of evaluation to prove the points the study makes or presents any argument that supports the reason of not including additional languages? Is these four languages enough to reach a conclusion about the general nature of cross-lingual transfer and debiasing, if yes, then why?
>
> We thank the reviewer for these interesting remarks. As we think these two comments are related, we provide a combined answer to both remarks.
>
> Cross-lingual debiasing is a very challenging and complex topic, with little research available. The debiasing techniques we are testing are made for English and research has mostly focused on this language. Research also mostly focuses on gender bias, neglecting other forms of bias, which already increases complexity in a monolingual context, so even more in a cross-lingual context. We acknowledge the keen observation that the languages included in the analysis are latin-script languages. However, we carefully selected  these four languages for our short paper format to **balance adding diversity and not overreaching conclusions that can be made with debiasing techniques made for English**. We included languages from **similar cultures as English** and cover both **Roman and Germanic languages**. Moreover, the chosen languages differ in the amount of data that was used to train mBERT as shown by Wu & Dredze (2020).  English is the highest resource language. French and German show similar amounts of resources, and is then followed by Dutch.  Hence, these **languages represent a diverse set for initial cross-lingual transferability analysis**. Furthermore, this offers a good first step to **stimulate research into this area** and show its importance.
>
> Including **non-Latin languages** is a very interesting idea that triggered our research curiosity that includes similar challenges as were present for some languages in our study due to the lack of an available dataset. However, including these languages **requires in-depth analysis that reaches beyond the short paper format**, since we do not wish to provide premature insights in an ethically sensitive research domain.
>
> Nevertheless, we **extended the evaluation setup** to show the results when debiasing in Russian by translating samples of the CrowS-Pairs dataset, despite the cultural differences for which extended analysis is necessary.  As expected, debiasing in Russian does not help to debias English in mBERT. For the other three languages, we could say that in general, Russian can be used to debias French, German, and Dutch in mBERT. However, **for Russian, the debiasing is not effective**.  The results suggest that this is due to overcompensating when debiasing gender bias. However, this requires **more in-depth analysis**, as observed variation can also be due to cultural differences, thereby potentially requiring an alternative experimental set-up. Given the ethical intricacies inherent to this research domain, we refrain from presenting these preliminary findings, as this requires in-depth analysis that goes beyond the scope of the short paper format.  Once again, thanks for this really interesting suggestion!
>
> | Evaluation language | mBERT | INLP  | SenDeb | DR*   | CDA   | DO    |
> |---------------------|-------|-------|--------|-------|-------|-------|
> | **EN**                  | 6.11  | 10.19 | 7.5    | 7.22  | 10.09 | 9.17  |
> | **FR**                  | 11.11 | 10.93 | 12.78  | 10.83 | 10.74 | 8.98  |
> | **DE**                  | 9.49  | 7.93  | 8.98   | 7.59  | 10.64 | 11.37 |
> | **NL**                  | 17.66 | 15.05 | 16.53  | 16.53 | 17.09 | 16.08 |
> | **RU**                  | 14.60 | 15.63 | 15.16  | 15.43 | 12.28 | 14.05 |
>
> |       |     | mBERT | INLP  | SenDeb | DR*    | CDA   | DO    |
> |-----|-----|-------|-------|--------|--------|-------|-------|
> | **EN**  |   | 51.11 | 51.85 | 51.94  | 51.67* | 58.80 | 52.31 |
> |    |Gender  |   49.17 |   45.83 |   50     |   50.83  |   48.61 |   49.72 |
> |   |Race | 44.17 | 43.06 | 45     | -      | 58.89 | 41.11 |
> |   |Religion | 60.00 | 66.67 | 60.83  | -      | 68.89 | 66.11 |
> | **FR**   |      | 60.56 | 55.19 | 61.11  | 59.72* | 59.26 | 57.69 |
> |   |Gender  | 50.83 | 43.89 | 47.50  | 48.33  | 52.50 | 56.11 |
> |   |Race | 59.17 | 53.06 | 62.50  | -      | 60.83 | 50.28 |
> |   |Religion | 71.67 | 68.61 | 73.33  | -      | 64.44 | 66.67 |
> | **DE**  |       | 59.21 | 55.57 | 58.13  | 56.45* | 57.61 | 56.46 |
> |   |Gender  | 61.39 | 48.73 | 54.77  | 53.10  | 56.91 | 54.00 |
> |   |Race | 57.07 | 62.41 | 58.78  | -      | 48.43 | 47.88 |
> |   |Religion | 59.17 | 55.56 | 60.83  | -      | 67.50 | 67.50 |
> | **NL**  |       | 67.66 | 64.87 | 64.58  | 64.58* | 66.34 | 65.89 |
> |   |Gender  | 56.32 | 52.66 | 47.07  | 47.07  | 50.97 | 54.35 |
> |   |Race | 65.83 | 61.94 | 65.83  | -      | 70.28 | 64.72 |
> |   |Religion | 80.83 | 80    | 80.83  | -      | 77.78 | 78.61 |
> | **RU**  |       | 63.18 | 61.41 | 60.95  | 61.24* | 59.20 | 61.88 |
> |   |Gender  | 52.03 | 45.06 | 45.34  | 46.22  | 45.93 | 47.02 |
> |   |Race | 79.17 | 78.89 | 78.33  | -      | 77.50 | 81.11 |
> |   |Religion | 58.33 | 60.28 | 59.17  | -      | 54.17 | 57.50 |
>
> >R3B: Also, to reach a conclusion about low-resource language, I feel more experiments need to be done on other true low-resource languages such as not seen by mbert languages.
>
> We thank the reviewer for this correct remark, we can indeed solely address the languages that have been investigated in our study. Therefore, we agree that more experiments have to be executed on other low-resource languages and encourage research into this direction with our paper. To adequately address this, we modified the sentences mentioned below.
>
> **Actions taken:** We changed the sentences on lines 18-21 as follows (changes in bold): *We also find that debiasing techniques with additional pretraining exhibit enhanced cross-lingual effectiveness **for the languages included in the analyses**, particularly in lower-resource languages.*, Lines 271-273: *Therefore we conclude that, **for our set of languages**, these techniques are most effective when applied to low-resource languages.* Lines  305-308: *Additionally, we find that, **for the studied languages**, debiasing with the lowest resource language is effective for techniques involving an additional training step (CDA and DO).*
> **References:** Shijie Wu and Mark Dredze. 2020. Are All Languages Created Equal in Multilingual BERT?. In Proceedings of the 5th Workshop on Representation Learning for NLP, pages 120–130, Online. Association for Computational Linguistics.

---

### Official Review · Reviewer_4hPj · 2023-08-03

**Soundness:** 2

**Excitement:**

3: Ambivalent: It has merits (e.g., it reports state-of-the-art results, the idea is nice), but there are key weaknesses (e.g., it describes incremental work), and it can significantly benefit from another round of revision. However, I won't object to accepting it if my co-reviewers champion it.

**Paper Topic And Main Contributions:**

Edit following author response:
I think that the author response helped clarify some details. Thank you very much!
I think that a dedicated methodology section with the details mentioned here.
I raised my score, but still am missing crucial details - most notably how were the translations obtained? I reread the paper and couldn't find additional details. Were they automatic? manual? Either case would require more elaboration on how this was done. What system / human translated the sentences? what was the quality of the translations?


-


Popular debiasing techniques are applied on mBERT on the English tokens, and the effect is tested on other languages.

I had a hard time understanding many of the details in the paper. The experimental setup boils down to the following sentence: "We debias mBERT using language X and evaluating it on language Y" (Line 195). I'm not sure what this means exactly or how to reproduce the results, furthermore, I couldn't understand how the results on languages other than English were obtained, since the paper itself mentions that the dataset used isn't available in other languages (other than French?).

I think that adding more details about the exact debiasing process, and which datasets are used is crucial for making the paper clearer. Following, I think that an analysis of the results would help shed light on what exactly changes when applying debiasing cross-lingualy, e.g., via a manual error analysis.

**Reasons To Accept:**

* The topic and idea of the paper is interesting and may benefit debiasing efforts in low resource languages.

**Reasons To Reject:**

* Many details missing - how was the debiasing carried out exactly, how were the evaluations carried out on the different languages.

**Reproducibility:**

3: Could reproduce the results with some difficulty. The settings of parameters are underspecified or subjectively determined; the training/evaluation data are not widely available.

**Reviewer Confidence:**

3: Pretty sure, but there's a chance I missed something. Although I have a good feel for this area in general, I did not carefully check the paper's details, e.g., the math, experimental design, or novelty.

---

> ### Author Rebuttal · Authors · 2023-08-29
>
> We thank you for the comments and your perspective on our research and as per your request elaborated  on the experimental setup and model evaluation conducted in the paper.
> > R2A: The experimental setup boils down to the following sentence: "We debias mBERT using language X and evaluating it on language Y" (Line 195). I'm not sure what this means exactly or how to reproduce the results
>
> Our paper investigates the **transferability of debiasing techniques across different languages within a multilingual model, mBERT**. We thoroughly investigate the effectiveness of these techniques across four languages: English, French, German, and Dutch. Our findings demonstrate the feasibility of  cross-lingual transfer of debiasing  and show promising results. More specifically, we debias the model  **using one language and subsequently evaluate its performance in another language**. We did this **for all possible combinations of the set of languages** within our research scope. Our paper highlights an important research problem as most research into bias mitigation has been done in a monolingual context, and mostly for English.
>
> **Actions taken:** Thanks to the reviewer comment, we revised our paper to improve the clarity. An example of this is adding the following sentence for extra clarification regarding the experimental setup: *This means that we debiased the model using one language and then evaluated the results on another language, and this for all different experimental languages.*
>
> > R2B: furthermore, I couldn't understand how the results on languages other than English were obtained, since the paper itself mentions that the dataset used isn't available in other languages (other than French?).
>
> As noted by the reviewer and mentioned in our paper, there is no other translated version available of the CrowS-Pairs dataset except for French. Therefore, we have **indicated in the paper that we have translated samples of the dataset** for the other respective languages. Moreover, we do wish to point out that **all datasets including translations are provided in the supplementary materials** and therefore contribute to the reproducibility of our research.
> **Actions taken:** We added extra clarification about our experimental setup to line 111: *Therefore, we used three samples of the full dataset and translated them to the respective language to evaluate our experiments.*
>
> > R2C: Following, I think that an analysis of the results would help shed light on what exactly changes when applying debiasing cross-lingualy, e.g., via a manual error analysis
>
> Thank you for the suggestion to manually investigate the model results. While we believe this is interesting to **distill potential linguistic patterns** that impact the bias prediction results, we do not agree with the premise that this manual error analysis will *“shed light on what exactly changes when applying debiasing cross-lingualy”*. During the evaluation, we compare the differences between the predicted probabilities for two tokens. Hence, the evaluation metrics capture the changes in predicted probabilities when applying the debiasing techniques cross-lingually.
>
> Nevertheless, we acknowledge that the paper benefits from **manually investigating examples where cross-lingual bias mitigation is most and least effective**. We randomly sample 10 sentences for each pair of debiasing and evaluation language, where the predicted token probabilities deviated most/least. We include a short analysis of selected samples below.
>
> Upon analyzing the best-performing sentences, we observe a variety of results present, with gender and race predominately present. **The worst-performing sentences are primarily associated to the religion bias category.** This is in line with the findings in Tables 7- 10, where often the bias for religion shows limited decrease in bias. For all different techniques, we find that more than 70% of the 100 worst-performing sentences are part of the religion category. Therefore, the gender category greatly influences the final result of the debiasing technique. That is why, when looking further into the results of Tables 1, 4, 5, and 6 and Figure 2, we find that often the reason for a technique to not perform well is that it overcompensates the bias for either gender or race when debiasing. Overcompensation occurs when a model is too much debiased, shifting the bias direction to the opposite direction.

---

### Official Review · Reviewer_pZuG · 2023-08-04

**Typos Grammar Style And Presentation Improvements:** Line 78
**Soundness:** 3

**Excitement:**

3: Ambivalent: It has merits (e.g., it reports state-of-the-art results, the idea is nice), but there are key weaknesses (e.g., it describes incremental work), and it can significantly benefit from another round of revision. However, I won't object to accepting it if my co-reviewers champion it.

**Paper Topic And Main Contributions:**

The paper studies the effectiveness of existing debiasing techniques in multilingual models.  Using BERT and translating the CrosS-Pairs dataset, the authors apply Counterfactual Data Augmentation (CDA), Dropout Regularization (DO), SentenceDebias (SenDeb), Iterative Nullspace Projection (INLP), and DensRay (DR) debiasing methods for English, French, German, and Dutch languages. The authors evaluate the debiasing methods' performance under a zero-shot setting, where they are applied to a language while being evaluated in other languages. The proposed results suggest that the debiasing techniques are effective in addressing social biases across the selected languages. The exceptional case is English, where applying debiasing methods increases the bias in mBERT.

**Questions For The Authors:**

NA.

**Reasons To Accept:**

The paper studies an important research problem in NLP. While most of the bias analysis and debiasing methods have been done in the monolingual area and mostly in English, this is important extend the research to other languages.

**Reasons To Reject:**

Although the question is clear and the results to some extent support it, there is still room to improve the analysis and understanding of bias in multilingual models. For example, looking at individual debiasing methods' performance in Table 1 and the appendix show that not all of them are effective in debiasing across languages. Moreover, the interesting results in Figure 2 can benefit from further analysis. Based on the results for some languages (e.g., NL), using another language (e.g., DE) is more effective in decreasing the bias.

**Reproducibility:**

4: Could mostly reproduce the results, but there may be some variation because of sample variance or minor variations in their interpretation of the protocol or method.

**Reviewer Confidence:**

4: Quite sure. I tried to check the important points carefully. It's unlikely, though conceivable, that I missed something that should affect my ratings.

---

> ### Author Rebuttal · Authors · 2023-08-29
>
> We thank you for your time and valuable feedback.
> We agree that this paper covers an important research problem in NLP, as most research into debiasing has been done in a monolingual setting (e.g, English). Below, we elaborate on your comments and feedback.
>
> >**R1A: There is still room to improve the analysis and understanding of bias in multilingual models. For example, looking at individual debiasing methods' performance in Table 1 and the appendix show that not all of them are effective in debiasing across languages**
>
> We agree that the paper benefits from further analysis providing additional insights into cross-lingual debiasing. Therefore, we include further analyses into the debiasing methods that are not effectively debiasing in the camera-ready version as per below.
> Due to the almost perfect bias score for English in the mBERT, debiasing techniques tend to increase the bias score. Ahn and Oh (2021) confirm this by proposing mBERT as debiasing technique. However, there are also other techniques that do not effectively debias the results. Many of these techniques show bad performance because of **overcompensation** after debiasing the model. We assume that an unbiased model has an equal chance of preferring the biased sentence over the unbiased sentence.  However, when the debiasing technique overcompensates the bias, it makes the prediction of the unbiased sentence more likely than the biased one. We saw this phenomenon occurring in many of the bad-performing techniques shown in Table 1:
> * When debiasing mBERT using **INLP with English and Dutch as debiasing languages and French as evaluation language**, we see that overall bias does not decrease due to gender bias being overcompensated.
> * When debiasing mBERT using **INLP with German as the debiasing language and evaluating the results on German**, we see again that the bias for gender is overcompensated. On top of that, we also find that when debiasing race, the bias increases.
> * When debiasing mBERT using **INLP with French as the debiasing language and evaluating the results in German**, we also find an overcompensation of the gender bias category, combined with an increased bias for religion.
> * When debiasing mBERT using **SenDeb and DR with French as the debiasing language and evaluating the results in French**, we find overcompensation for the gender bias category.
> * When debiasing mBERT using **CDA with French as the debiasing language and evaluating the results in German**, we find an overcompensation of the bias present in the race category.
> * When debiasing mBERT using **DO with English as the debiasing language and evaluating the results in German**, we also find overcompensation of the bias present in the race category and a small increase in bias for religion.
>
> **Actions taken:** The additional analysis is added to the camera-ready version of the paper.
> **References:** Jaimeen Ahn and Alice Oh. 2021. Mitigating language-dependent ethnic bias in BERT. In Proceedings of the 2021 Conference on Empirical Methods in Natural Language Processing, pages 533–549, Online and Punta Cana, Dominican Republic. Association for Computational Linguistics.
>
> >R1B: Moreover, the interesting results in Figure 2 can benefit from further analysis. Based on the results for some languages (e.g., NL), using another language (e.g., DE) is more effective in decreasing the bias.
>
> To further clarify the figure, we added a **table** to show the best debiasing language for every evaluation language in our study.  This table is discussed in detail in the paper. We highlight the most important findings below.
>
> | **Evaluation Language** | **Best Debiasing Language** | **Worst Debiasing Language**  |
> |---------------------|-------------------------|---------------------------|
> | **English**             | Dutch                   | French                    |
> | **French**              | German                  | French                    |
> | **German**              | German                  | French                    |
> | **Dutch**               | French                  | English                   |
>
> * **Dutch is shown to be the best-performing debiasing language for English** because this debiasing language overcompensates gender bias the least.
> * In general, using the **same debiasing as evaluation language results in overcompensation of the bias**, turning around the bias direction. Therefore it is the best debiasing language is often not the same as the evaluation language. However, German has the highest bias score for gender before debiasing, and therefore strong debiasing is beneficial.
> * **German is the best-performing debiasing language for French** since it shows the best performance on all different evaluation sets. Moreover, it also shows less overcompensation for the gender bias present in the model than other languages such as Dutch
> * **French is the worst-performing debiasing language for all languages except for Dutch, where it is the best-performing one**. We find that when evaluating in French, the gender bias is overcompensated. For English, both racial and gender bias are overcompensated. The German evaluation shows lower overall performance because of already two ineffective methods (INLP and CDA), which were also due to overcompensating racial bias. Finally, for Dutch, we find that debiasing with French overcompensated gender bias less than Dutch and therefore is the best-performing method.
>
> We believe that these results are influenced by the fact that both German and French have a grammatical gender distinction, which could have an impact on debiasing gender to a greater extent. This grammatical gender distinction is not the case for English and Dutch.
>
> **Actions taken:** The table and additional explanations are added to the camera-ready version of the paper.
>
> >**R1C: Line 78: an extra '.' .**
>
> We thank the reviewer for pointing this out and have deleted the extra '.' in the camera-ready version of the paper.

---

### Meta-Review · Area_Chair_NsCF · 2023-09-19

**Recommendation:** 3

**Metareview:**

This paper investigates the transferability of several debiasing techniques across languages, when applied to multilingual models.

The reviewers appreciated that this work investigates debiasing in multilingual models, particularly since model debiasing work has centered on English. However, the reviewers felt that the paper would benefit from additional experiments (i.e., on a larger and more diverse set of languages), and more clarity, particular in the analysis.

---

### Decision · Program_Chairs · 2023-10-07

**Decision:**

Accept-Main

**Comment:**

This paper investigates the transferability of several debiasing techniques across languages, when applied to multilingual models.

The reviewers appreciated that this work investigates debiasing in multilingual models, particularly since model debiasing work has centered on English. However, the reviewers felt that the paper would benefit from additional experiments (i.e., on a larger and more diverse set of languages), and more clarity, particular in the analysis.